# Delineating the Role of *Aedes aegypti* ABC Transporter Gene Family during Mosquito Development and Arboviral Infection via Transcriptome Analyses

**DOI:** 10.3390/pathogens10091127

**Published:** 2021-09-02

**Authors:** Vikas Kumar, Shilpi Garg, Lalita Gupta, Kuldeep Gupta, Cheikh Tidiane Diagne, Dorothée Missé, Julien Pompon, Sanjeev Kumar, Vishal Saxena

**Affiliations:** 1Department of Biological Sciences, Birla Institute of Technology and Science (BITS), Pilani Campus, Pilani 333031, India or p2016019@pilani.bits-pilani.ac.in (V.K.); shilpi@pilani.bits-pilani.ac.in (S.G.); 2Department of Zoology, Chaudhary Bansi Lal University, Bhiwani 127021, India or lalita.zoology@cblu.ac.in; 3Department of Radiology and Radiological Sciences, School of Medicine, 470 Cancer Research Building-II, Johns Hopkins University, Baltimore, MD 21218, USA; kgupta14@jhmi.edu; 4MIVEGEC, Univ. Montpellier, IRD, CNRS, 34394 Montpellier, France; tidiane.diagne@ird.fr (C.T.D.); dorothee.misse@ird.fr (D.M.); julien.pompon@ird.fr (J.P.); 5Department of Biotechnology, Chaudhary Bansi Lal University, Bhiwani 127021, India

**Keywords:** *Aedes aegypti*, ABC transporters, immunity, arboviruses, dengue virus

## Abstract

*Aedes aegypti* acts as a vector for several arboviral diseases that impose a major socio-economic burden. Moreover, the absence of a vaccine against these diseases and drug resistance in mosquitoes necessitates the development of new control strategies for vector-borne diseases. ABC transporters that play a vital role in immunity and other cellular processes in different organisms may act as non-canonical immune molecules against arboviruses, however, their role in mosquito immunity remains unexplored. This study comprehensively analyzed various genetic features of putative ABC transporters and classified them into A-H subfamilies based on their evolutionary relationships. Existing RNA-sequencing data analysis indicated higher expression of cytosolic ABC transporter genes (E & F Subfamily) throughout the mosquito development, while members of other subfamilies exhibited tissue and time-specific expression. Furthermore, comparative gene expression analysis from the microarray dataset of mosquito infected with dengue, yellow fever and West Nile viruses revealed 31 commonly expressed ABC transporters suggesting a potentially conserved transcriptomic signature of arboviral infection. Among these, only a few transporters of ABCA, ABCC and ABCF subfamily were upregulated, while most were downregulated. This indicates the possible involvement of ABC transporters in mosquito immunity.

## 1. Introduction

The *Aedes aegypti* (*Ae. aegypti*) mosquito transmits a wide range of arboviruses such as dengue (DENV), Chikungunya (CHIKV), West Nile (WNV), yellow fever (YFV) and Zika virus (ZIKV) that are known to cause severe health problems in human. Among these, DENV is one of the most dangerous infections, which inflicts over 390 million individuals annually with nearly half of the world population at the risk of infection [1,2]. The large human population, expanding urbanization, a significant increase in trade as well as tourism, and climate change are the major contributing factors to augment the disease transmission [3]. Various efforts to eliminate or control the mosquito vectors have faced a significant setback due to the development of insecticide resistance and the unavailability of effective disease transmission-blocking vaccines. These facts substantiate a need to understand the mechanisms and factors responsible for the disease transmission and to develop strategies for its control.

The interaction of viruses with the mosquito system is initiated just after their ingestion. They spatially and sequentially interact with different mosquito organs; gut, being the first organ of encounter and salivary glands as the last organ of interaction. Although these organs mount an array of immune responses against pathogens, still, the virus somehow manages to escape the mosquito immunity [4,5,6]. Many studies have identified critical mosquito molecules that either inhibit or support virus propagation and survival. Notably, the mosquito immune signaling pathways such as Toll, IMD and JAK-STAT are reported to be activated against a wide range of pathogens and are negatively regulated by Cactus, Caspar and PIAS, respectively [7,8,9,10]. A recent report also revealed the broad antiviral role of *Ae. aegypti* JNK pathway [11]. In addition, the mosquito machinery that cleaves viral RNA through RNA interference (RNAi) also exhibits an antiviral response [12,13].

Identification and characterization of non-canonical or novel immune molecules, which may regulate pathogen development might offer new insights to interrupt the mosquito cycle of blood-borne pathogens [14]. ABC transporters, the transmembrane molecular pumps, which utilize ATP to translocate a variety of molecules across the biological membrane have been shown to play an important immune role in some insects [15,16]. For example, RNAi-mediated silencing of an ABC transporter sulphonylurea receptor (Sur), which is part of a potassium channel, makes *Drosophila* hypersensitive to viral infections [15]. In addition, some of the ABCC subfamily transporters are also reported to exhibit a strong negative correlation with *Plasmodium* infection in *Anopheles gambiae* [17,18].

ABC transporters share a typical architectural organization comprising highly conserved hydrophilic nucleotide-binding domains (NBDs) and less conserved hydrophobic transmembrane domains (TMDs) [19]. NBDs, also known as ATP-binding cassettes, are located at the cytoplasmic side of the cell membrane and hydrolyze ATP to generate translocation driving energy. They contain various conserved signature motifs like Walker A (GXXGXGK(S/T)), Walker B (φφφφD, here φ denotes a hydrophobic residue), Q-loop, D-loop, and H-motif, as well as C-motif (LSGGQ). The transmembrane domains (TMDs) form the translocation pathway and provide specificity to a substrate [20,21,22]. Based on the number or presence/absence of NBDs or TMDs, the ABC transporters have been characterized as full, half or cytosolic transporters. The full ABC transporter contains four functional domains, two NBDs and two TMDs encoded by a single gene. The members that contain one TMD and one NBD, are called half transporters and are functional as homo—or heterodimers [23]. The members that lack the TMDs are termed cytosolic transporters and are mainly regulatory in function. Insect ABC transporters superfamily is further subdivided into eight subfamilies (ABCA to ABCH), based on the phylogenetic relationship among their NBDs [22,24]. 

In this study, we identified and comprehensively analyzed the ABC transporters superfamily from *Ae. aegypti* genome assembly 5 (AaegL5) [25] through characterization of their conserved domains, structural topology, chromosomal location, and phylogenetic relationship. Further, the expression patterns of these transporters were also analyzed using the available transcriptome datasets to delineate their functions during mosquito development and post arboviral infections [26,27]. The transcriptomic alterations were validated by quantitative real-time PCR (qPCR) for a few representative genes.

## 2. Results

### 2.1. Genomic Location and Curated Details of Ae. aegypti ABC Transporters

We have identified a total of 59 putative ABC transporter genes from *Ae. aegypti* genome assembly 5 (AaegL5) [25], which is in contrast to previous studies performed by Lu et al. and Figueira-Mansur et al. [26,28] that reported 69 and 53 ABC transporters, respectively from *Ae. aegypti* genome. These reports included long non-coding RNA, uncharacterized protein, and pseudogenes, which is now curated in the recently updated genome assembly (AaegL5) [25]. Our analyses revealed that among 59 transporter genes, 47 genes have expression sequence tags (ESTs) support, while the remaining 12 genes are without ESTs, however, they all showed the evidence of expression in at least one of the analyzed transcriptomic datasets (Table 1).

Based on the *An. gambiae* transporters classification [27], *Ae. aegypti* ABC transporters were also classified into ABCA to ABCH subfamilies (Figure 1A,B). These transporters were further classified into full, half, or cytosolic transporters based on their structural topology (Figure 1B). The ABCA, ABCB, and ABCC subfamilies consist of ten, one, and fifteen full transporter genes, respectively; while the ABCD, ABCG, and ABCH subfamilies have only half transporters. The ABCE and ABCF subfamilies contain cytosolic transporters and do not have TMDs (Figure 1A,B). 

#### Chromosome Mapping of ABC Transporter Genes 

Chromosomal mapping revealed that the identified 59 ABC transporter genes are located on three different chromosomes (Figure 1C). Among these, 23.73% of total ABC transporter genes (14 genes) are located on chromosome 1. The genes located on chromosome 1 belong to ABCA (1/11 members), ABCC (5/16 members), ABCD (2/2 members), ABCG (4/18 members) and ABCH (2/3 members) subfamilies. 42.37% of total ABC transporter genes (25 genes) are located on chromosome 2. These include clusters of ABCC (6/16 members) and ABCG (10/18 members) located on the long arm, while a cluster of ABCA (4/11 members), ABCB (3/5 members), ABCC (1/16 members) and ABCG (1/18 members) is located on the short arm of chromosome 2. Chromosome 3 has remaining 33.9% of total ABC transporter genes (20 genes) from different ABC subfamilies, including ABCA (6/11 members), ABCB (2/5 members), ABCC (4/16 members), ABCE (1/1 members), ABCF (all 3 members), ABCG (3/18 members), and ABCH (1/3 members). 

In the comparison of *Ae. aegypti* ABC transporters with other reported insect species (such as *Drosophila melanogaster* [24], *An. gambiae* [27], *Bombyx mori* [30], *Tribolium castaneum* [31], *Anopheles sinensis* [32] and *Culex quinquefasciatus* [26]), it was observed that most of the ABC transporters belong to the ABCA, ABCC, and ABCG subfamilies. The ABCE is the smallest subfamily with only a single member (Table 2). The ABCD, ABCF and ABCH had a similar number of ABC transporters in all insects. 

### 2.2. Phylogenetic Analyses and Characteristics of Ae. aegypti ABC Transporters

Phylogenetic trees were built to categorize the *Ae. aegypti* ABC transporters into different subfamilies (Figure 2) and to understand the evolutionary pattern of *Ae. aegypti* ABC transporters with orthologues from other insects (Appendix A). When compared to the reference genome of *An. gambiae*, we observed gene duplication and addition of new members indicating genome expansion in *Ae. aegypti*. The gene duplication was evident in ABCA, ABCC and ABCG subfamilies. Maximum subfamilies exhibited similar branch length indicating common evolutionary distance. 

#### 2.2.1. ABCA Subfamily 

*Ae. aegypti* has eleven putative members in the ABCA subfamily, all of which are full transporters except *AaeABCA7.2*, which is a half transporter. A 1:1 orthologous relation was observed among most of the members of *Aae*ABCA subfamily with those from other insects analyzed in this study. There were duplications of transporters such as *AaeABCA2.1* and *AaeABCA2.2* as well as *AaeABCA7.1* and *AaeABCA7.2*, that are homologous to *An. gambiae* AGAP006380 (*ABCA2*) and AGAP001523 (*ABCA7*), respectively. Interestingly, the duplication of *AaeABCA7* transporter was observed only in *Ae. aegypti*. These duplicated genes formed a common clade with 100% bootstrap support (Appendix A). An additional member of the ABCA subfamily, *AaeABCA10* was identified in the *Aedes* genome, which was not observed in *An. gambiae* and other insects. Although this gene branches within the ABCA phylogeny group, it appeared as a distant clade. We could not identify any ortholog in the AaegL5 for *ABCA3* transporter of *An. gambiae* and *Cx. quinquefasciatus*. The transporters from *Ae. aegypti* ABCA subfamily varied from 1773 to 7298 amino acids in length. Among the *Aedes* ABC transporters, the ABCA subfamily members clustered adjacent to ABCG subfamily in a common clade. A detailed phylogeny of ABCA subfamily members revealed that *Drosophila* ABCA transporters are present at the base of each clade followed by *An. gambiae* homologs and *Aedes* ABCA transporters branching at the end of each clade. 

#### 2.2.2. ABCB Subfamily

We identified five *Ae. aegypti* ABCB subfamily members that share a 1:1 relationship with other insect orthologs. Interestingly, no gene duplication and addition/deletion could be identified in this subfamily. *Ae. aegypti* ABCB subfamily formed a common clade with ABCC and ABCD subfamilies (Figure 2). All of these are members of mitochondrial ABC systems and are known to play roles in the iron metabolism and Fe/S protein precursors transport [34]. Upon sequence analysis, *AaeABCB1* showed sequence conservation with *AaeABCB4* while *AaeABCB3* showed similarity with *AaeABCB5*. Upregulation of *Ae. aegypti ABCB4* is reported in a pyrethroid-resistant strain [35]. It is of note that all the members of *Ae. aegypti ABCB* subfamily are half transporters except *ABCB2*. *Ae. aegypti ABCB2* is homologous to *D. melanogaster* biochemical defense genes (*DmMdr49*, *DmMdr50*, and *DmMdr65*) and *An. gambiae* ABCB2 (AGAP005639) (Appendix A). *Aae*ABCB2 is a full transporter known as P-glycoprotein, which is linked to insecticide resistance in *D. melanogaster* and other arthropods [22,36] and has been reported to exhibit a detoxification effect against the insecticide temephos in *Ae. aegypti* larvae [16]. 

#### 2.2.3. ABCC Subfamily

*Ae. aegypti* ABCC transporters constitute the second-largest ABC subfamily, with 15 full transporters and one-half transporter. All these transporters exhibit 1:1 homology with *An. gambiae* ABCC members. Interestingly, *AaeABCC4* has undergone duplication into *AaeABCC4*.1 and *AaeABCC4*.2, and both are located on the opposite strands of the same chromosome. These two duplicates were observed to share high nucleotide similarity (80%) with identical introns-exons architecture. We could not identify *ABCC11* and *ABCC15* members of this subfamily, which are present in *An. gambiae*, indicating their absence in *Ae. aegypti* genome. In the phylogenetic tree, *AaeABCC17* formed a cluster with *Aedes albopictus* and *Culex* orthologs (Appendix A), but surprisingly it branched away from the rest of the ABCC subfamily cluster and is present adjacent to *AaeABCE1* (Figure 2). The appearance of *An. gambiae* transporter genes at the base of all clusters indicated their possible early origin when compared to *Aedes* and *Culex*.

#### 2.2.4. ABCD Subfamily 

The ABCD subfamily of *Ae. aegypti* consists of only two half transporters. This subfamily also has a true orthologous relationship with *An. gambiae* counterparts. Our phylogenetic analyses revealed that *AaeABCD1* clusters with other insect orthologs with 100% bootstrap support, while *AaeABCD2* clusters with its orthologs from other insects with only 71% bootstrap (Appendix A). These ABC proteins are reported to be involved in fatty acid transport into the peroxisome in insects [37]. 

#### 2.2.5. ABCE and ABCF Subfamilies

Both these subfamilies also show 1:1 orthology with ABC transporters from other insects. *Aae*ABCE has only a single member, while *Aae*ABCF has three members. Members of both subfamilies show NBD-NBD topology and are devoid of TMDs. Thus, these transporters may not perform any transport function but might play a fundamental role in cellular processes such as translation initiation, elongation, termination, and ribosome biosynthesis etc. as suggested in other organisms [38,39]. *AaeABCE1* transporter clusters along with the *Aae*ABCA and *Aae*ABCH clades; while *Aae*ABCF subfamily members clusters with *Aae*ABCB, *Aae*ABCD and *Aae*ABCC clades indicating their close evolutionary origin (Figure 2). 

#### 2.2.6. ABCG and ABCH Subfamilies

With eighteen putative members, the ABCG subfamily is the largest transporters subfamily in *Ae. aegypti*. All its members are half-transporters, like most eukaryotic counterparts except for plants and fungi (which have full transporters). All the *Ae. aegypti* ABCG subfamily members possess a typical reverse domain architecture (NBD-TMD). Among themselves, all the members show high conservation forming a distinct clade upon phylogeny. Most of the AaeABCG transporters show 1:1 orthology with other insects except for *An. gambiae ABCG6*, which was not identified in *Ae. aegypti*. *AaeABCG19*/20 (AAEL026976) is orthologous to both *An. gambiae* AGAP009471 (*ABCG19*) and AGAP009472 (*ABCG20*). We were also able to identify the *AaeABCG22* as reported previously [26], which is not identified in *An. gambiae* (Appendix A). The *Aae*ABCG cluster falls in a clade along with *Aae*ABCA and *Aae*ABCH subfamilies with ~90% bootstrap, indicating their common evolutionary origin (Figure 2). The *Aae*ABCH subfamily was first identified in *D. melanogaster* and is exclusively reported in insects with no ortholog in mammals, plants, or yeast. Three members of the ABCH subfamily were identified in *Ae. aegypti* that have a similar topology of NBDs and TMDs to the *Aae*ABCG transporters, although their function is not yet known.

#### 2.2.7. ABCJ Subfamily

An additional ABC transporter J subfamily was also reported in *Ae. aegypti* genome, which included relative ABC-ATPases like SMC, Rad50 and MutS [28]. The members of this subfamily are soluble in nature and have a role in DNA repair processes (MutS and Rad50 proteins), and chromosome condensation (SMC) Among these, while Rad50 has a well-conserved LSGG motif, Walker A and B motifs for nucleotide-binding [40], SMCs have more degenerate signature motif and minimal Walker A and B motifs [41]. MutS do not contain signature motif and classical Walker-A [42]. There exist different arguments for the inclusion of these proteins in the ABC transporter superfamily [42,43]. We have also identified an ATP-sensitive K^+^ channel in *Ae. aegypti,* similar to other organisms [44] that also possess features like ABC transporters. Thus, more detailed analyses are required for the inclusion of the J subfamily in the ABC superfamily.

### 2.3. Expression Profiling of ABC Transporters in Different Developmental Stages of Ae. aegypti 

To delineate and distinguish the importance of ABC transporters, we analyzed their expression profile in different developmental stages (eggs, larvae, pupae and adults) and various organs (carcass and gonads) of mosquito based on the transcriptome data from Akbari et al. [45]. The K-means clustering of all *Aae*ABC transporters, based on their mRNA expression profile was performed (Figure 3) as discussed in Section 4.4. The ABC transporter genes showing high expression patterns (1.5–2.5 log_10_ expression) from embryo to adult stages, as well as in blood-fed adults, formed a separate cluster (Cluster 1, Figure 3) and included the cytoplasmic subfamily members *AaeABCE1*, *AaeABCF2* and *AaeABCF3*. 

Among the 59 ABC transporters, the expression was only detected for 51 transporters (in at least one of the developmental stages), except for four members each from *Aae*ABCA (*AaeABCA1, AaeABCA2.1, AaeABCA2.2* & *AaeABCA10* members) and *Aae*ABCC (*AaeABCC4.1, AaeABCC4.2, AaeABCC5 & AaeABCC*6 members) subfamilies. These eight members fall in a distinct cluster (Cluster 2, Figure 3) along with other members exhibiting reduced or minimal expression. Most of the genes in this cluster showed varied expression during different stages except *AaeABCA7.1* and *AaeABCB5* genes that showed a decent expression throughout mosquito development. The genes showing varied expression during specific stages include *AaeABCA6*, which showed a high level of expression in NBF ovaries; *AaeABCA4, AaeABCG3* and *AaeABCH1* that were expressed mainly in carcasses of males with minimal expression in female carcasses (BF and NBF). Of these genes, *AaeABCH1* showed good expression during larval and pupal development also while *AaeABCG3* exhibits minimal expression during these stages. *AaeABCA8* and *AaeABCC9* showed expression in both male and female carcasses as well as during embryonic development. Few genes that did not show expression in carcass are *AaeABCG13* and *AaeABCG14*, however, they exhibit a high level of expression during embryonic development.

The genes that showed a decent to high-level expression throughout the mosquito development formed a separate cluster (Cluster 3, Figure 3). However, among these genes, *AaeABCC10*, *AaeABCG2* and *AaeABCG11* showed minimal expression in ovaries of blood-fed (BF) females.

### 2.4. Validating the Expression Pattern of Selected ABC Transporter Genes in Different Developmental Stages of Ae. aegypti 

To further validate the expression of randomly selected ABC transporter genes (*AaeABCC9*, *AaeABCF2*, *AaeABCG11* and *AaeABCG13*) in different developmental stages of mosquito i.e., eggs, larval stages, pupae and adult males or females, we performed qPCR (Figure 4) using gene-specific primers (primer sequences are given in Appendix A).

The qPCR data of the selected genes (Figure 4) revealed a similar pattern when compared to the RNA-seq data (Figure 3). As mentioned previously, *AaeABCC9* showed maximum expression in male carcass which is also observed in the qPCR data for this gene. *AaeABCF2* gene shows expression throughout the development of mosquitoes with maximal expression in 1st instar larvae and adult stage (in both male and female) which is also evident in the qPCR data. *AaeABCG13* expression was minimal during different developmental stages as per the RNA seq data and a similar pattern was observed after qPCR validation.

### 2.5. Expression Profile of Ae. aegypti ABC Transporters upon Arboviral-Infections 

During arboviral infection in the mosquito, the virus must disseminate from the midgut (MG) through the body (CC) to the salivary gland (SG). Therefore, the gene expression is likely to be altered in different organs at various time points. To determine the spatial and sequential expression of *Ae. aegypti* ABC transporters, we analyzed datasets available from microarray experiments on three arboviral infections viz. DENV, YFV and WNV [7,46,47,48]. Infection kinetics was generated as per virus progression in the mosquito, beginning from its ingestion to readiness for subsequent transmission [on Day1, D2, D7, D10 and D14 post-infection (dpi)] in the whole mosquito as well as in its different organs including CC, MG and SG. 

Most of the *Ae. aegypti* ABC transporters had a significant differential expression in the whole body of mosquito up to 7 dpi (YFV, WNV and DENV). Our analyses detected the mRNA expression for 54 genes (91.52%) in at least one of the arboviral infections at 1–7 dpi. Out of these 48 genes (81.35%) showed differential expression in at least one of the arboviral infections (Figure 5). Forty-three genes were differentially expressed during YFV infection, 36 genes during WNV and 43 genes during DENV infection. Among these, 31 genes were commonly expressed during all three infections. *AaeABCA7*.2 and *AaeABCC4*.2 were exclusively expressed during the early infection of YFV while *AaeABCA8*, *AaeABCB3* and *AaeABCF1* were exclusively downregulated 7 dpi in DENV (Figure 5A). Almost all members of *Aae*ABCB, *Aae*ABCG and *Aae*ABCH subfamilies showed continuous downregulation upon any of the three arboviral infections. In contrast few members of subfamilies *Aae*ABCA (*Aae*ABCA2.1, *Aae*ABCA6), *Aae*ABCC (*Aae*ABCC7, *Aae*ABCC10, *Aae*ABCC14) and *Aae*ABCF (*Aae*ABCF2) were upregulated during arboviral infection. *AaeABCC14* was continuously upregulated on all-time points of YFV, WNV and DENV infection [7]. 

Further to analyze the DENV infection kinetics we also investigated other available reports for 10 dpi and 14 dpi of DENV [46,47]. Contrary to the significantly differential expression of all ABC transporters 1–7 dpi, only two genes from the ABCG subfamily were significantly modulated at 10 and 14 dpi, where *AaeABCG3* gene was downregulated in the carcass at 10 dpi of DENV [46] and *AaeABCG11* was upregulated in the carcass and salivary gland on 14 dpi of DENV infection [47]. Several ABC transporters were also found downregulated in the mosquito cell lines Aag2 upon DENV infection, similar to the in vivo experiments [48].

### 2.6. Validating the Expression of ABC Transporter Genes in DENV2 Virus-Infected Ae. aegypti 

To validate the above transcriptome data (Figure 5), we randomly selected five ABC transporter genes (*AaeABCC13*, *AaeABCE1*, *AaeABCF2*, *AaeABCG11* and *AaeABCG13*) and assessed their expression levels through qPCR. For the same, RNA was isolated from DENV2 infected *Ae. aegypti* as well as uninfected blood-fed mosquitoes at different time points. The actin gene was used as an internal control. The qPCR expression pattern of these genes revealed a similar trend as reported for the microarray data (Figure 6). 

*AaeABCC13* exhibited no differential regulation at early DENV2 infection (at 1 and 2 dpi) but was observed to be downregulated at 7 dpi, similar to the microarray reports. *AaeABCE1* and *AaeABCF2* transporters exhibited similar expression patterns post-DENV2 infection; both genes being significantly upregulated at the onset of infection (1 dpi), with subsequent downregulation as the infection progressed. In contrast, a continuous downregulation was observed for *AaeABCG11* and *AaeABCG13* genes post-DENV2 infection in accordance with the microarray data.

### 2.7. Analysis of Promoters and Transcription Factors Binding Site (TFBS) in The Regulatory Region of Virus-Induced ABC Transporters

To understand the transcriptional regulation of ABC transporter genes having probable roles in mosquito immunity during arboviral infections, the binding sites for transcription factors (TFBS) were predicted in the 5′ UTR sequences of a few representative ABC transporter genes. We could identify the binding sites for STAT, Rel, Cnc, Bgb, Sch and Ets in these ABC transporter genes (Figure 7A), where the 5′ UTR sequences of each gene showed binding sites for single/multiple TFs. Further to correlate the possible transcriptional regulation of ABC transporter genes by these factors, we examined the expression patterns of these TFs in arboviral infected mosquito transcriptome [7]. Rel1 was observed to be upregulated in DENV infection with no significant change in WNV and YFV, however, Rel2 and Bgb showed downregulation in all arboviral infections. STAT was upregulated only at 2 dpi in DENV while Cnc variants showed upregulation at variable stages in all three arboviral infections. Interestingly, another transcription factor, Sch commonly associated with fatty-acid synthesis, was found to be upregulated throughout the three arboviral infections (till 7 dpi) (Figure 7B). The presence of immune mechanism related TFBS in 5′ UTR of these ABC transporter genes indicate their possible role during mosquito immunity. However, to identify the role of individual TF with specific ABC transporters, a time-dependent transcriptome profiling is required.

## 3. Discussion

The involvement of ABC transporters in multiple functions including lipid transport, multidrug resistance (MDR), transport of xenobiotics, chemotherapeutic drugs, and immunity, indicates their accessibility to a wide range of substrates and pathways [49,50]. However, the probable role of ABC transporters in mosquito immune mechanisms post-arboviral infections has not been elucidated. The identification of ABC transporters in different mosquito genomes reveals interesting traits of these proteins in fundamental cellular and housekeeping processes. Many of these proteins, even when belonging to the same subfamily often exhibit different functions, and proteins from different subfamilies may show related features. As detailed in earlier reports, and as observed in our phylogenetic analysis, the ABC transporters are highly conserved across various mosquito genus which are common hosts/vectors for different viruses including DENV, WNV, YFV, CHIKV etc. and protozoan parasites [7]. The conservation across these protein subfamilies in their functional domains may be attributed to their structural topology. 

In the current study, the comparative sequence and phylogenetic analysis of 59 putative *Ae. aegypti* ABC transporters suggested a common evolutionary ancestral origin with other insect ABC transporters. There is a high level of conservation of *Aae*ABC transporters with orthologs from different insects, including the dipterans: secondary vector for DENV (*Ae. albopictus*), fruit fly (*D. melanogaster*), malaria vector mosquito (*An. gambiae*) and *Cx. quinquefasciatus*, etc. The structural architecture of *Aae*ABC members is also highly conserved with their respective homologues from other insects, plants, and humans. Interestingly, the *Aae*ABC transporters subfamily E remains the smallest like other insects and humans; while, subfamilies C and G have maximum members throughout insect genomes. Gene duplications was observed for *AaeABCA2* (*AaeABCA2.1* and *AaeABCA2.2*), *AaeABCA7* (*AaeABCA7.1* and *AaeABCA7.2*) and *AaeABCC4* (*AaeABCC4.1* and *AaeABCC4.2*) while there were certain gene deletions (*AaeABCA3*, *AaeABCC11*, *AaeABCC15*, and *AaeABCG6*) and gene additions (*AaeABCA10*, *AaeABCC16*, *AaeABCC17*, *AaeABCG21* and *AaeABCG22*) when compared to *An. gambiae* transporters. The ABC transporters of *Ae. aegypti* still seems less in number when compared to *Cx. quinquefasciatus* which has the highest number of ABC transporters genes among the reported insect genomes; however, it has many pseudogenes and is still under curation. The *Ae. aegypti* ABCB subfamily members seem parallel with *Anopheles* and *Drosophila* having sequence homology of ~30% with the human TAP proteins, which are involved in protein translocations for antigen processing [50]. As an extension to the defense mechanism, *DmMdr65* (orthologue of *AaeABCB2*) has been reported to impart chemical protection to the insect brain and acts as the efflux transporter at the blood-brain barrier [51].

To ensure the functions of *Ae. aegypti* ABC transporters, we performed an in-depth comparative analyses of these genes through mosquito developmental stages and at different time-points post arboviral (YFV, WNV or DENV) infections. A few of the ABC transporters exhibited contrasting expression patterns during developmental stages and post arboviral infections, as observed for *AaeABCA2.1* and *AaeABCA6* genes. These transporters were upregulated during viral infections through 1 dpi to 7 dpi, but their expression was scanty or minimal during mosquito development stages. *AaeABCA4* was also seen upregulated at 7 dpi on DENV infection while it had null expression during development stages [7,45]. This indicates the pro-viral role of these transporters. In contrast, *Ae. aegypti ABCA8*, *ABCB3* and *ABCC3* genes were downregulated at 7 dpi, but they had relatively higher expression during development stages. Similarly, *Ae. aegypti ABCA9*, *ABCB2*, *ABCC1*, *ABCC9*, *ABCG8*, *ABCG10*, *ABCG11*, *ABCG13*, *ABCG15* to *ABCG20* transporters had optimum to high expression profile during development to adult stages, while they were mostly downregulated during arboviral infections [7,45]. Our analysis revealed commonly expressed genes in three arboviral infections suggesting a potentially conserved transcriptomic signature of arboviral infection regardless of the type of viral infection. To investigate the transcriptomic alteration of these transporters, we analyzed and compared the high throughput sequencing data with our qPCR expression profiles and found correlations in the expression of representative genes like *Ae. aegypti ABCG11* and *ABCG13.* These genes showed high expression during adult stages of mosquito development similar to expression patterns observed during data analysis from Akbari et al. [45] Similarly, down-regulatory patterns were observed for these transporters during arboviral infection in both analyses (qPCR expression as well as bioinformatics analysis) [7]. These observations indicate the importance of these transporters during mosquito development while their downregulation on arboviral infection indicates the possible modulation of mosquito immune mechanisms by viruses.

The reported role of various ABC transporters in different systems motivated us to further investigate their expression patterns. The downregulation of the peroxisomal ABC transporter *AaeABCD2* on 2 dpi and 7 dpi of arbovirus suggested the modulation of peroxisomal machinery by the viruses for the establishment of infection in the host, as reported in the mammalian system [52]. Similarly, the downregulation of ABC transporters from subfamily C and G has been reported to play a role in the transportation of bacteriocin, glutathione conjugates and lipid-derived eicosanoids and is known to be involved in insect immune response to infection, indicating their role in antiviral immunity [18]. Further *ABCE1* role has been detailed during viral infection and anti-apoptosis in humans [39]. *AaeABCE1* showed an increased expression during mosquito development and at 1 dpi of DENV, however, it is downregulated as the infection progresses further, confirming its potential role in viral infection. 

The arboviral infected *Ae. aegypti* salivary gland transcriptome analysis revealed upregulation of several ABC transporters from A, C, G and H (Appendix A) suggesting their possible role in arboviral transmission [11,47]. Our analysis also predicted putative transcription factors for viral-induced *Ae. aegypti* ABC transporters, one of such is *AaeABCC14*, which is regulated by Rel, STAT, Cnc and Sch. Sch is involved in SREBP signaling, which is critical for viral propagation [53,54] and *AaeABCC14* is a putative sterol transporter. Both Sch and *AaeABCC14* were upregulated in the analyzed transcriptomic profiles (Figure 5B and Figure 7B), indicating the possible regulatory involvement of Sch for *AaeABCC14* gene. 

These findings necessitate a detailed gene-specific study to establish the functions of above listed ABC transporters in mosquito immune development against arboviral infections. The results of this research will lay the foundation for functional research and genetic evolution analysis. The discovery of novel/non-canonical genes related to mosquito development and immunity is important for accelerating the transmission-blocking research to control mosquito-borne diseases.

## 4. Materials and Methods

### 4.1. Ae. aegypti Mosquito Rearing

*Ae. aegypti* mosquitoes were reared in a semi-natural insectarium at 27 ± 1 °C, 80% relative humidity (RH) and 12 h light:dark cycle as described before [55,56]. Larvae were fed on fish food (Gold Tokyo, Ahmedabad, India) while the adult mosquitoes were regularly fed on 10% sucrose solution *ad libitum*. For colony propagation, four to five-day-old females were fed on anesthetized mice and their eggs were collected in moist cups. The mice were maintained in the central animal facility and all the procedures were approved by the animal ethics committee (IAEC/RES/28/7). 

### 4.2. Identification and Classification of Ae. aegypti ABC Transporters

To identify putative *Ae. aegypti* ABC transporters, protein sequences of already reported fifty-five *An. gambiae* ABC transporters [24,27] were retrieved from VectorBase [57] and subjected to BLASTp search against *Ae. aegypti* genome assembly 5 (AaegL5) [58]. The retrieved hits from *Ae. aegypti* genome that exhibited blast e-value <10^−6^ were further screened for ABC transporter specific signature domains through the conserved domain database (CDD) tool at NCBI [59]. Furthermore, the profile hidden markov model was adopted to characterize the NBDs and TMDs in these identified putative *Ae. aegypti* ABC transporters [29].

Finally, the presence of ABC transporter sequences in *Ae. aegypti* was confirmed through BLASTp search in its genome using already identified ABC transporters from *D. melanogaster* and *Cx. quinquefasciatus* as query. In addition, tBLASTn search was also performed using the protein sequences of *An. gambiae* ABC transporters to rule out the possibility of mis/un-annotated or new ABC transporter genes in *Ae. aegypti* genome assembly 5 (AaegL5).

The putative *Ae. aegypti* ABC transporters were classified into AaeABCA to AaeABCH subfamilies through multiple sequence alignment (Clustal Omega), phylogenetic tree reconstruction and orthologous relationship as reported before [26,27,60]. The nomenclature adopted for *Ae. aegypti* ABC transporters was based on their homology with *An. gambiae* ABC transporters [27]. The *Ae. aegypti* ABC transporters that exhibited one to one orthology with *An. gambiae* ABC transporters were assigned the same nomenclature. However, when more than one *Ae. aegypti* ABC transporter exhibited orthology with a single ortholog in *An. gambiae*, the *Ae. aegypti* ABC transporter with maximum similarity was suffixed with X.1 while the one with less similarity was named X.2 and so on (where X is the gene nomenclature corresponding to *An. gambiae*). Interestingly, some of the *Ae. aegypti* ABC transporters exhibited no orthology with *An. gambiae* transporters and thus, they were named based on homology with their paralogs in *Ae. aegypti*.

### 4.3. Phylogenetic Analysis of Ae. aegypti ABC Transporters 

To understand the evolutionary relationship of putative *Ae. aegypti* ABC transporters, we performed their phylogenetic analyses with *An. gambiae*, *Cx. quinquefasciatus*, *D. melanogaster*, and *Ae. albopictus* ABC transporters, using the neighbor-joining (NJ) method in MEGAX program [33]. The evolutionary distances were computed using the Poisson correction method and represented as the unit of number of amino acid substitutions per site. All positions with less than 95% site coverage were eliminated, i.e., fewer than 5% alignment gaps, missing data, and ambiguous bases were only allowed at any position (pairwise deletion). The tree topology confidence was assessed through bootstrap analysis using 1000 replicates.

### 4.4. Transcriptomic Data Analysis of Ae. aegypti ABC Transporters 

To analyze the expression patterns of ABC transporters during mosquito development, all 42 RNA-seq libraries initially described by Akbari et al. [45] and already remapped to the *Ae. aegypti* AaegL5 assembly was retrieved from VectorBase (https://vectorbase.org/vectorbase/app/search/transcript/GenesByRNASeqEvidence#GenesByRNASeqaaegLVP_AGWG_SRP026319_ebi_rnaSeq_RSRC, accessed on 15 July 2020) and analyzed using R program. First, the raw counts per gene were calculated from RNA-seq data using “featureCounts” [60]. These raw counts were further converted to the fragments per kilobase per million mapped reads (FPKM) using “countToFPKM” module. The obtained FPKM values were log10-transformed (by adding 1 to it) before generating a heatmap by “ComplexHeatmap’’ using K-means clustering on rows as reported before [61]. Differential expression analysis was performed for selected groups using edgeR [62,63].

Furthermore, to analyze the expression of *Ae. aegypti* ABC transporters during different arboviral infections like WNV, YFV or DENV, we constructed a single non-redundant gene list of these transporters based on prior AaegL1.2 and AaegL3.3 and current AaegL5.1 annotations. We analyzed the corresponding gene expression from the publicly available microarray datasets [7,46,47,48], wherein the reported probe sets were used to identify genes in new genome assembly through BLASTn. Genes identified with a significant *p*-value <0.05 were considered for expression analysis. The log_2_ fold change in the expression was used to construct a heatmap using Morpheus software (https://software.broadinstitute.org/morpheus, latest accessed on 29 October 2020).

### 4.5. Sample Collection and Analyzing the Expression Profile of Representative ABC Transporter Genes Using qPCR

Different developmental stages of *Ae. aegypti* such as eggs, larvae (first to fourth-instar), pupae, adult males or females were collected in RNAlater and stored at −80 °C. Total RNA was isolated from these samples using RNeasy mini kit (Qiagen, Hilden, Germany) with slight modification by adding 30 μL β-mercaptoethanol (2-ME) per 1 mL RLT buffer. First-strand cDNA was synthesized using QuantiTect reverse transcription kit (Qiagen, Hilden, Germany) according to manufacturer’s instructions followed by qPCR using SYBR green supermix in an IQ5 multicolor detection system (Bio-Rad, Hercules, CA, USA), where the mRNA for mosquito ribosomal protein subunit S6 gene was used as a normalization control [56]. The primer sets of representative genes used for expression analyses are mentioned in Appendix A. Following PCR cycle parameters were used, initial denaturation at 95 °C for 3 min, 35 cycles of 10 s at 95 °C, 40 s at 55 °C, and 1 min at 72 °C. The fluorescence readings were taken after each cycle. A final extension at 72 °C for 10 min was performed followed by a melting curve analysis.

### 4.6. Viral Infection, Sample Collection and Expression Analysis of Representative ABC Transporter Genes

Three-to-five-day old female mosquitoes were starved overnight and offered an infectious blood meal containing rabbit blood washed with 1× PBS and supplemented with 10 mM ATP (Thermo Scientific, Waltham, MA, USA), 10% fetal bovine serum (Sigma, Kawasaki, Japan), 7.5% Sodium bicarbonate and DENV2 virus titer suspended in RPMI media (Gibco, Scotland, United Kingdom) maintained at 1 × 10^6^ pfu/mL. Control mosquitoes were fed with the same blood meal composition except for the virus. Fully engorged females were selected and kept in a cage with 10% sucrose solution *ad libitum* in incubation at standard rearing conditions. Sham treated control and infected mosquitoes were collected at 1, 2 and 7 dpi.

Total RNA was extracted from the collected samples using E.Z.N.A. Total RNA kit I (Omega Bio-tek, Norcross, GA, USA) and isolated RNA was treated with DNase using Turbo DNA-free kit (Thermo Fisher Scientific), and reverse transcribed by iScript cDNA synthesis kit (Biorad, Hercules, CA, USA). Gene expression was quantified in LightCycler 96 qPCR system (Roche, Mannheim, Germany) using EvaGreen Master Mix (Euromedex, Lyon, France) following standard protocol, using gene-specific primers as detailed in Appendix A. Mosquito actin gene expression was used as a normalization control [64]. The reactions were performed using the following cycle conditions: an initial 95 °C for 5 min, 40 cycles of 95 °C for 15 s, 60 °C for 20 s and 72 °C for 20 s followed by a melting curve analysis. Fold changes were calculated using the ^ΔΔ^Ct method as described before [65]. 

All the data were expressed as mean + standard deviation (SD). Statistical significance was analyzed by one-way ANOVA with Tukey’s multiple comparison test for the differential gene expression using GraphPad Prism 8.0 (GraphPad Software, La Jolla, CA, USA). The data with a *p*-value < 0.05 were considered significant.

### 4.7. Identification of Promoter and Transcription Factor Binding Sites in ABC Transporter Genes

To identify promoters, the 5′ upstream sequence of each *Ae. aegypti* ABC transporter gene was analyzed using the online NPP2.2 tool (http://www.fruitfly.org/seq_tools/promoter.html, accessed on 29 March 2021), (organism selected = Eukaryotes and Minimum promoter score = 0.8) [66]. The upstream 5′UTR sequence to promoter of selected ABC transporter genes was analyzed with online JASPAR software (http://jaspar.genereg.net/search?q=&collection=CORE&tax_group=insects, accessed on 31 March 2021) to predict the transcription factor binding (TFB) sites. Here the insect core database was selected, using *Drosophila* as a reference organism and a similarity threshold of 80% [67].

## Figures and Tables

**Figure 1 pathogens-10-01127-f001:**
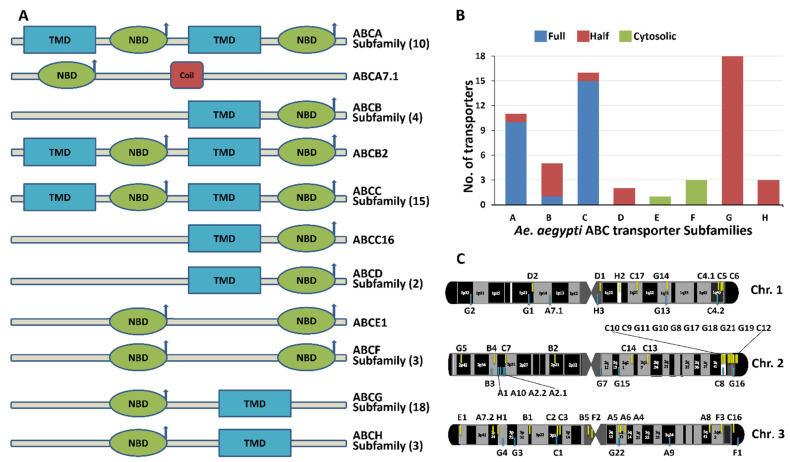
(**A**) Structural topology of *Ae. aegypti* ABC transporters. The subfamilies of the ABC transporters are marked on the right-hand side of the structure and the total number of transporters with similar structures in the subfamily are denoted in parenthesis. The arrow denotes the active site (ATPase activity). The transporter *ABCA7.1* exclusively has the coiled-coil domain (denoted by the red square). The position of NBD and TMD domains of each *Ae. aegypti* ABC transporter was identified using the EBI program “phmmer” [29]. (**B**) Subfamily-wise distribution of full, half and cytosolic ABC transporters represented by blue, red and green color, respectively. (**C**) Chromosomal distribution of *Ae. aegypti* ABC transporter genes. The physical location of ABC transporter genes is represented in yellow (sense strand) and blue (antisense strand), and the corresponding gene name, in short, is mentioned on the top or bottom side of the individual chromosome.

**Figure 2 pathogens-10-01127-f002:**
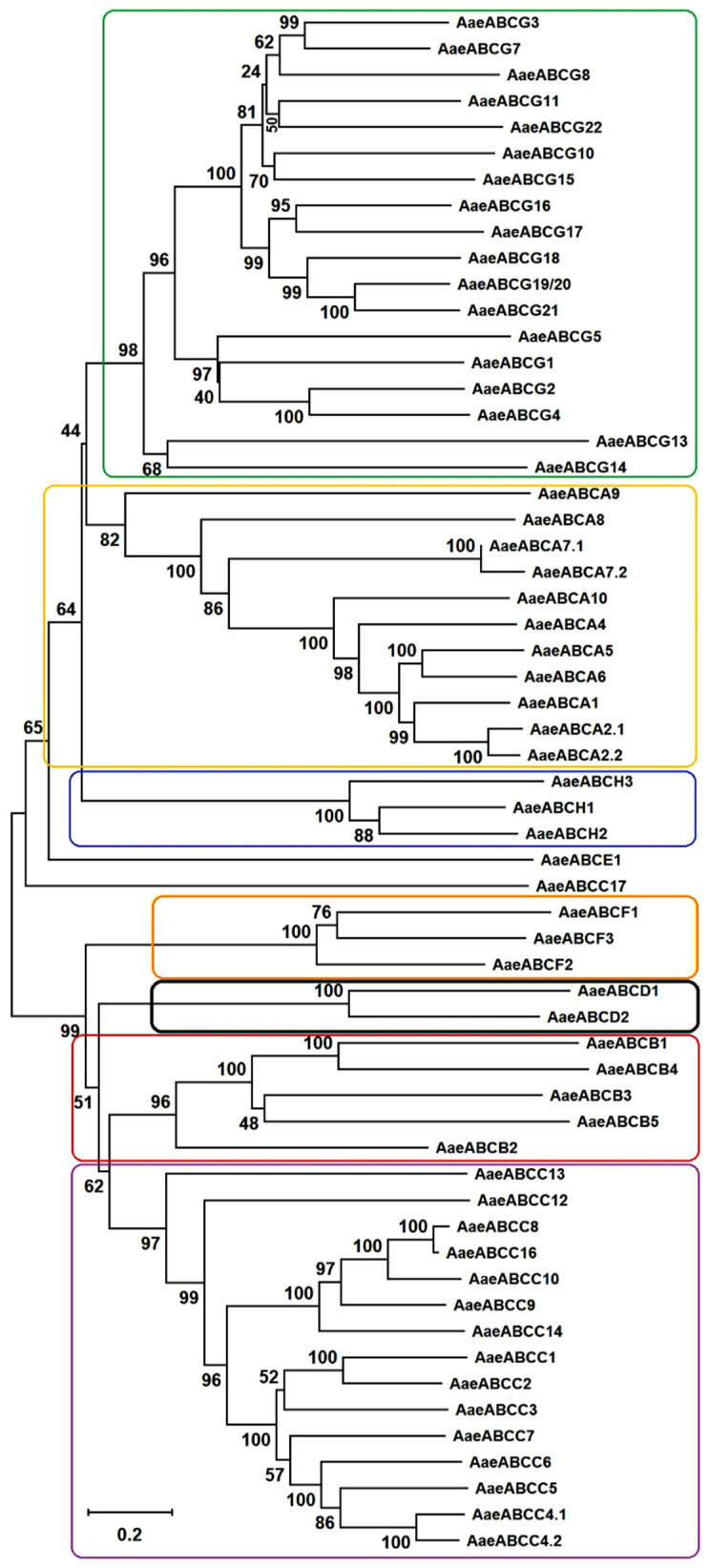
Phylogenetic relationship of *Ae. aegypti* ABC transporters inferred with the Neighbor-Joining method using MEGA X [33]. The bootstrap value calculated from 1000 replicates is marked on each corresponding node. The members of each subfamily are enclosed in a different color box except for *ABCE1* (a single member of the subfamily) and *ABCC17* (falling apart from its other subfamily members).

**Figure 3 pathogens-10-01127-f003:**
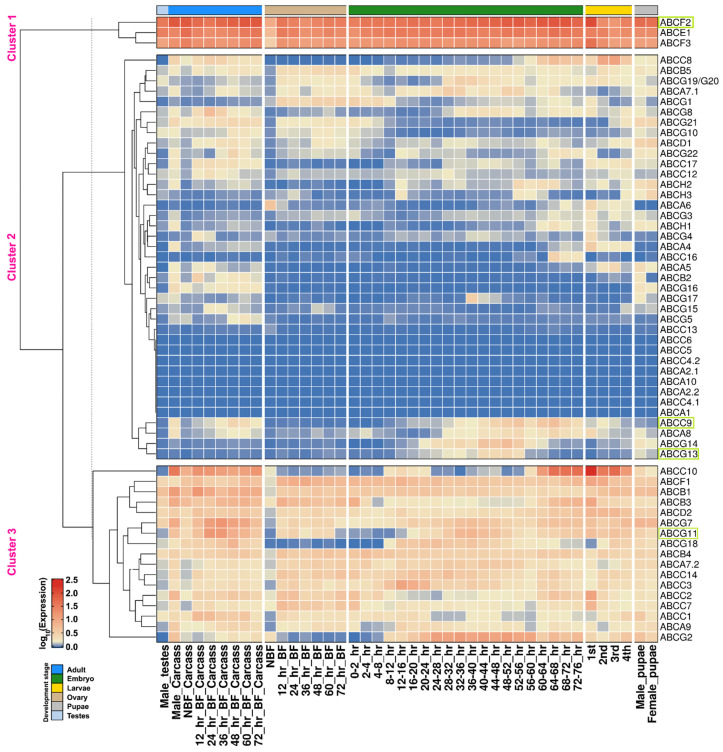
Expression profiling of *Ae. aegypti* ABC transporter transcripts in different developmental stages as well as at different time points after the blood-feeding using K-means clustering [45]. The Color scheme indicates the log_10_ FPKM expression value. NBF, non-blood-fed females; BF, blood-fed females; 1st to 4th instar stages of larvae. The expression pattern of few genes validated by qPCR is marked in boxes.

**Figure 4 pathogens-10-01127-f004:**
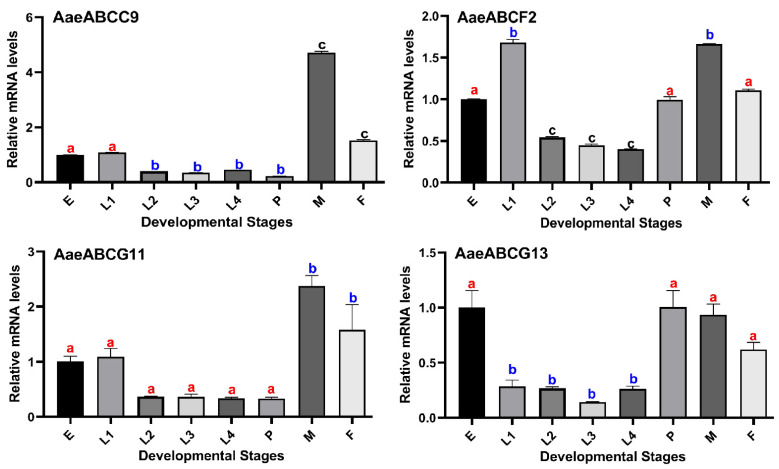
qPCR analysis of randomly selected ABC transporters in different developmental stages of *Ae. aegypti* mosquito. E = Eggs, L = Larvae (numbers indicate the instar of larvae L1 to L4), P = Pupae, M = Males and F = Females. All samples were collected from three biological replicates at each developmental stage, error bars indicate mean ± SD (*n* = 3). The relative mRNA levels were calculated as mentioned in Materials and Methods. The mRNA levels in eggs were considered as 1.0 in each analysis. Significant differences among relative mRNA levels are represented with different alphabets like “a, b, c”. The ‘a’ alphabet bearing bars have non-significant expression change to each other, while they have significant differences with bars bearing other alphabets (b & c).

**Figure 5 pathogens-10-01127-f005:**
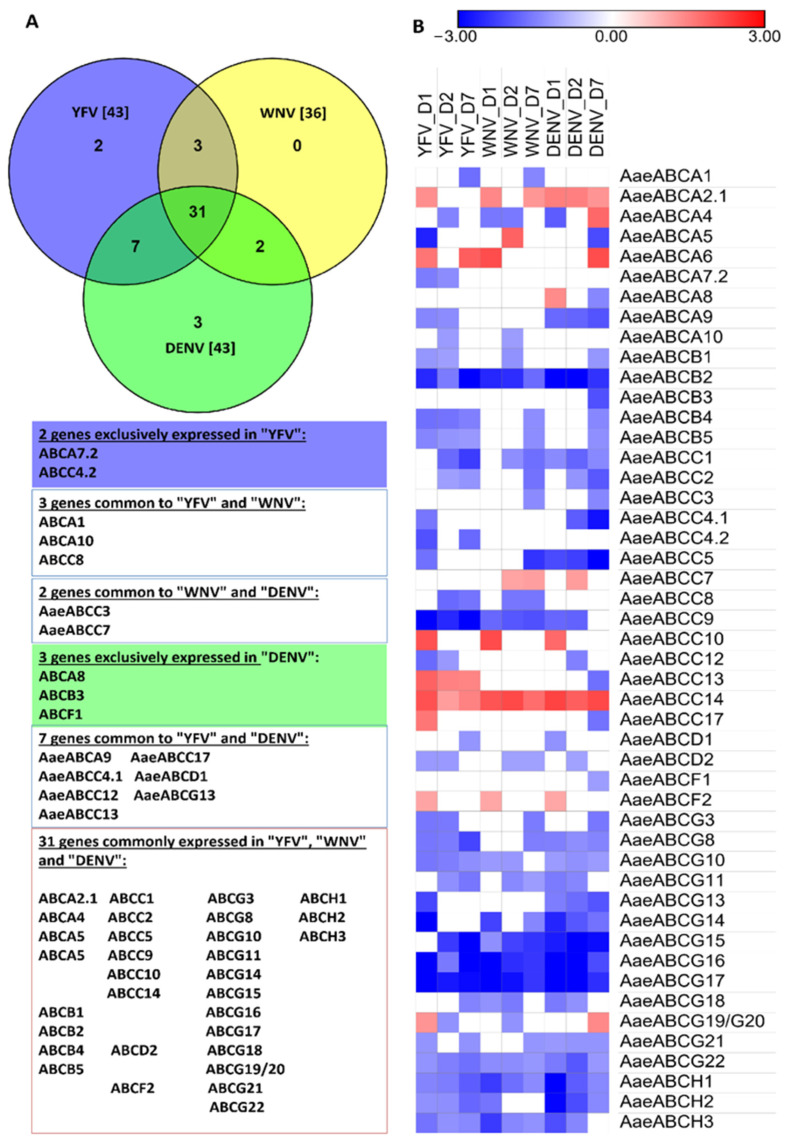
Analysis of *Ae. aegypti* ABC transporters expression profile in WNV 2741, DENV-2 New Guinea C or YFV Asibi strain infected mosquitoes. As described by Colpitts et al., 2011, virus was used at 6.5 logs per mL, and 0.5 µL was injected intra-thoracically per mosquito. The viral infection was confirmed by RT-qPCR [7]. (**A**) The Venn diagram represents the number of exclusively or commonly expressed genes post arboviral infection. The list of these genes is given at the bottom of the diagram. (**B**) Heatmap of log2 values of differentially regulated transcripts of Ae. aegypti ABC transporters at different days post virus infection [7]. Only the genes exhibiting a significant expression (*p* < 0.05) are represented here. The Color scheme indicates the log2 Fold-change expression of genes against controls at D1, D2, D7 (D = represents the day post-infection).

**Figure 6 pathogens-10-01127-f006:**
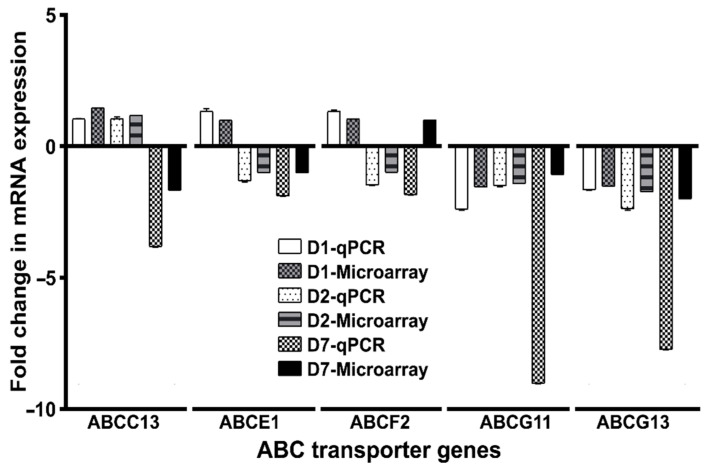
Comparative analysis of the microarray and qPCR expression data of a few representative ABC transporter genes (as mentioned on the x-axis) during DENV2 infection. The fold change in mRNA expression was calculated by ^ΔΔ^Ct method as described in Materials and Methods, using the actin gene as an internal control. All samples were collected from three biological replicates of each treatment (DENV2 infected blood— v/s uninfected blood-fed) at specified time intervals. Error bars indicate mean ± SD (*n* = 3). D1, D2 and D7 indicate the days post-viral infection. The sequence of the primers used in qPCR is given in Appendix A.

**Figure 7 pathogens-10-01127-f007:**
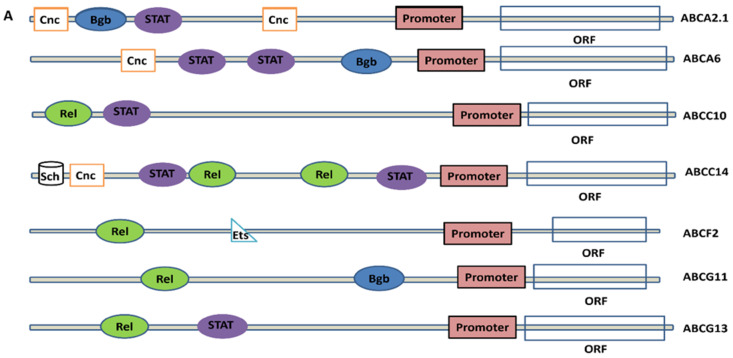
(**A**) Representation of putative transcription factors binding sites (TFBS) in the regulatory region of *Ae. aegypti* ABC transporter genes. ORF = Open reading frames, STAT = STAT binding site, Rel = Rel binding site, Bgb = Protein big brother binding site, Cnc = cap’n’ collar binding site, Ets = Ets binding site and Sch = Schlank binding site. (**B**) Heatmap showing differential expression of transcription factor encoding genes in arboviral infected *Ae. aegypti* mosquitoes [7]. Only genes showing a significant *p*-value (*p* < 0.05) are shown here. D1, D2, D7 represent the days post arboviral infection.

**Table 1 pathogens-10-01127-t001:** ABC transporters identified from *Ae. aegypti* genome assembly 5 (AaegL5) [25]. (+) = Sense strand; (−) = Antisense strand; Y—Yes, N—No.

ABC Transporter Gene	VectorBase ID	Chromosome: Physical Location (Strand)	No. of Exons	No. of Transcripts	Protein Length (AA)	Availability of EST (No. of ESTs)	Expression Status in	
							Mosquito Developmental Stages	Arboviral Infection
**ABCA Subfamily (11 Genes)**								
** *AaeABCA1* **	AAEL012698	2:94093797–94103717(−)	8	3	1641	Y (4)	N	Y
** *AaeABCA2.1* **	AAEL012702	2:94177008–94183151(−)	11	1	1636	Y (26)	N	Y
** *AaeABCA2.2* **	AAEL012700	2:94160180–94166404(−)	11	1	1645	N	N	Y
** *AaeABCA4* **	AAEL001938	3:227108694–227135343(+)	8	1	1673	Y (11)	Y	Y
** *AaeABCA5* **	AAEL008386	3:224734279–224779573(+)	9	2	1663	Y (5)	Y	Y
** *AaeABCA6* **	AAEL008384	3:224779816–224802105(+)	10	1	1657	Y (1)	Y	Y
** *AaeABCA7.1* **	AAEL017572	1:117261376–117263613(−)	2	1	590	N	Y	N
** *AaeABCA7.2* **	AAEL014699	3:51218604–51279389(+)	7	2	1635	Y (18)	Y	Y
** *AaeABCA8* **	AAEL021738	3:365856244–365899503(+)	12	2	1900	Y (3)	Y	Y
** *AaeABCA9* **	AAEL018040	3:322613800–322714818(−)	16	2	1987	Y (26)	Y	Y
** *AaeABCA10* **	AAEL012701	2:94115322–94145067(−)	8	1	1622	Y (6)	N	Y
**ABCB Subfamily (5 Genes)**								
** *AaeABCB1* **	AAEL008134	3:107404546–107431730(+)	5	1	848	Y (20)	Y	Y
** *AaeABCB2* **	AAEL010379	2:182523007–182607302(+)	11	3	1307	Y (2)	Y	Y
** *AaeABCB3* **	AAEL022941	2:78738879–78757036(−)	3	1	725	Y (10)	Y	Y
** *AaeABCB4* **	AAEL006717	2:85121437–85146605(+)	6	2	820	Y (9)	Y	Y
** *AaeABCB5* **	AAEL000434	3:187161644–187197938(+)	7	1	693	Y (8)	Y	Y
**ABCC Subfamily (16 Genes)**								
** *AaeABCC1* **	AAEL005918	3:138401412–138418395(−)	9	1	1312	Y (17)	Y	Y
** *AaeABCC2* **	AAEL025460	3:138444756–138480164(+)	9	4	1388	Y (13)	Y	Y
** *AaeABCC3* **	AAEL005929	3:138489236–138524582(+)	11	3	1419	Y (7)	Y	Y
** *AaeABCC4.1* **	AAEL012395	1:284711177–284717415(+)	5	2	1357	Y (9)	N	Y
** *AaeABCC4.2* **	AAEL019847	1:284693063–284698462(−)	6	1	1355	N	N	Y
** *AaeABCC5* **	AAEL023958	1:284737104–284863028(+)	13	1	1488	Y (4)	N	Y
** *AaeABCC6* **	AAEL027539	1:284869084–284903303(+)	11	3	1286	N	N	Y
** *AaeABCC7* **	AAEL018267	2:100234384–100334146(+)	7	6	1394	Y (23)	Y	Y
** *AaeABCC8* **	AAEL005045	2:445037808–445059189(−)	5	2	1508	N	Y	Y
** *AaeABCC9* **	AAEL005026	2:445021476–445037944(+)	5	1	1454	N	Y	Y
** *AaeABCC10* **	AAEL005043	2:444993656–444999015(+)	6	2	1522	Y (13)	Y	Y
** *AaeABCC12* **	AAEL020303	2:463313906–463367902(+)	5	1	1532	Y (2)	Y	Y
** *AaeABCC13* **	AAEL023524	2:312907331–313134735(+)	25	1	2101	Y (1)	Y	Y
** *AaeABCC14* **	AAEL004743	2:289194273–289251024(+)	15	15	1524	Y (27)	Y	Y
** *AaeABCC16* **	AAEL017209	3:403021800–403026639(+)	4	1	986	N	Y	N
** *AaeABCC17* **	AAEL015644	1:193810018–193840226(+)	9	1	1339	Y (10)	Y	Y
**ABCD Subfamily (2 Genes)**								
** *AaeABCD1* **	AAEL010047	1:160091832–160259983(+)	7	2	753	Y (14)	Y	Y
** *AaeABCD2* **	AAEL002913	1:108683838–108734858(+)	10	1	659	Y (27)	Y	Y
**ABCE Subfamily (1 Gene)**								
** *AaeABCE1* **	AAEL010059	3:11225514–11239622(+)	5	1	609	Y (66)	Y	Y
**ABCF Subfamily (3 Genes)**								
** *AaeABCF1* **	AAEL001101	3:406460452–406484817(−)	2	1	894	Y (16)	Y	Y
** *AaeABCF2* **	AAEL010977	3:190430154–190451030(+)	4	2	602	Y (97)	Y	Y
** *AaeABCF3* **	AAEL010359	3:372039893–372056253(+)	5	1	712	Y (52)	Y	Y
**ABCG Subfamily (18 Genes)**								
** *AaeABCG1* **	AAEL016999	1:107942728–108005678(−)	8	3	716	Y (17)	Y	N
** *AaeABCG2* **	AAEL021570	1:30489715–30492114(−)	4	1	687	N	Y	N
** *AaeABCG3* **	AAEL008138	3:79113447–79286804(−)	8	5	862	Y (15)	Y	Y
** *AaeABCG4* **	AAEL003703	3:65749828–65763092(−)	6	1	616	Y (1)	Y	Y
** *AaeABCG5* **	AAEL017188	2:24405169–24438842(+)	8	1	614	N	Y	N
** *AaeABCG7* **	AAEL008672	2:239556260–239647710(−)	7	2	699	Y (29)	Y	Y
** *AaeABCG8* **	AAEL019463	2:455430514–455514073(+)	7	1	723	Y (10)	Y	Y
** *AaeABCG10* **	AAEL027367	2:455272483–455402105(+)	11	7	726	Y (6)	Y	Y
** *AaeABCG11* **	AAEL008635	2:455184210–455236417(+)	9	1	676	Y (17)	Y	Y
** *AaeABCG13* **	AAEL022734	1:215452060–215689779(−)	10	5	888	Y (1)	Y	Y
** *AaeABCG14* **	AAEL027424	1:215707794–215774030(+)	6	1	602	Y (12)	Y	Y
** *AaeABCG15* **	AAEL019641	2:270847147–271027814(−)	12	3	604	Y (8)	Y	Y
** *AaeABCG16* **	AAEL008625	2:455514069–455540158(−)	5	1	606	N	Y	Y
** *AaeABCG17* **	AAEL008628	2:455557594–455572391(+)	6	2	598	N	Y	Y
** *AaeABCG18* **	AAEL008632	2:455591457–455609699(+)	6	3	607	Y (4)	Y	Y
** *AaeABCG19/ G20* **	AAEL026976	2:455663955–455680847(+)	7	3	599	N	Y	Y
** *AaeABCG21* **	AAEL008624	2:455615233–455650540(+)	6	5	593	Y (6)	Y	Y
** *AaeABCG22* **	AAEL027686	3:223544987–223689078(−)	8	1	612	Y (12)	Y	Y
**ABCH Subfamily (3 Genes)**								
** *AaeABCH1* **	AAEL005491	3:59371559–59533418(+)	8	5	814	Y (1)	Y	Y
** *AaeABCH2* **	AAEL018334	1:181178199–181420930(+)	11	5	814	Y (3)	Y	Y
** *AaeABCH3* **	AAEL014428	1:159149149–159331370(−)	12	3	727	Y (1)	Y	Y

**Table 2 pathogens-10-01127-t002:** Subfamily-wise gene number comparison of *Ae. aegypti* ABC transporters with other organisms.

ABC Transporter Subfamilies	*Homo* *sapiens*	*Drosophila* *melanogaster*	*Anopheles* *gambiae*	*Aedes* *aegypti*	*Culex* *quinquefasciatus*	*Tribolium* *castaneum*	*Bombyx mori*
**A**	12	10	9	11	10	10	6
**B**	11	8	5	5	5	6	8
**C**	12	14	14	16	18	35	15
**D**	4	2	2	2	2	2	2
**E**	1	1	1	1	1	1	1
**F**	3	3	3	3	3	3	3
**G**	5	15	18	18	28	13	13
**H**	0	3	3	3	3	3	3
**Total**	48	56	55	59	70	73	51

## Data Availability

The data presented in this study are available in Appendix A and it was taken from existed microarray and RNA-seq datasets under accessions: GSE28208, SRR8921123-8921132, SRP026319 [7,11,45].

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
