# Peer review of "Delineating the Role of Aedes aegypti ABC Transporter Gene Family during Mosquito Development and Arboviral Infection via Transcriptome Analyses"

_pathogens, 2021, doi:10.3390/pathogens10091127_

Round 1

Reviewer 1 Report

Kumar et al identified ABC transporter genes and analyzed their expression profiles according to published transcriptome and microarray data in Ae. aegypti. They also validated their expressions in different stages and their transcript expression changes upon arboviral expression by using qPCR. Here are my comments.

  • Most of the study in the manuscript was focusing on characterizing of ABC transporter genes in Ae. aegypti, and did not involve the functional studies such as loss- (knock out/down) and gain- (overexpression) function studies. In this case, the title is a little aggressive. I would like to substitute ‘functional’ with a more accurate word.
  • L 31-32: From the data presented in the manuscript, I am not convinced by ABC transporters were key players of mosquito immunity. Please consider to change it.
  • Figure 3. Please indicate the number of biological replicates. So do figure 4, 5, and 6.
  • Figure 5. Were all the ABC transporter gene expression data normalized to uninfected (only blood fed) mosquitoes? Please make it clear in the figure legend. It would be better that the viral titer and infection prevalence were included in the figures, and confirmed mosquitoes were infected by YFV, WNV or DENV in different tissues at different days post viral infection.
  • Figure 6. Was any statistical analysis performed on these data? Please indicate.
  • Line 533. Please write details about viral infection. Which methods were used to detect virus titers in mosquitoes? WNV and YFV information was missing here. Please indicate the virus origin and other details.

Author Response

Comments and Suggestions for Authors

Reviewer 1

Kumar et al identified ABC transporter genes and analyzed their expression profiles according to published transcriptome and microarray data in Ae. aegypti. They also validated their expressions in different stages and their transcript expression changes upon arboviral expression by using qPCR. Here are my comments.

  • Most of the study in the manuscript was focusing on characterizing of ABC transporter genes in Ae. aegypti, and did not involve the functional studies such as loss- (knock out/down) and gain- (overexpression) function studies. In this case, the title is a little aggressive. I would like to substitute ‘functional’ with a more accurate word.

Answer- As per reviewers’ suggestion, we have modified the title. Now the Updated title should be read as

“Delineating the role of Aedes aegypti ABC transporter gene family during mosquito development and arboviral infection via transcriptome analyses”.

  • L 31-32: From the data presented in the manuscript, I am not convinced by ABC transporters were key players of mosquito immunity. Please consider to change it.

Answer- Considering the reviewer's suggestion we have reframed the sentence as

“This indicates the possible involvement of ABC transporters in mosquito immunity”.

  • Figure 3. Please indicate the number of biological replicates. So do figure 4, 5, and 6.

Answer- Figures 3 and 5 were prepared based on the existing transcriptomic datasets (Akbari et al., 2013 and Colpitts et al., 2011, respectively)

Figure 3: We approached the author Prof. Omar S Akbari (Akbari et al., 2013) to obtain the information regarding biological replicates as enquired by the reviewer. The response of Prof. Akbari is following-

 “Each sample only had one replicate. Back then (in 2013) - RNAseq was very expensive and that was considered acceptable.” Therefore, the dataset analysed (figure 3) should be acceptable to the scientific community as justified above. In addition, the dataset provided by Akbari et al., 2013, has also been used in other different publications, for example, “Adelman, Zach N., and Kevin M. Myles. "The C-type lectin domain gene family in Aedes aegypti and their role in arbovirus infection." Viruses 10.7 (2018): 367.

Matthews, Benjamin J., et al. "Improved reference genome of Aedes aegypti informs arbovirus vector control." Nature 563.7732 (2018): 501-507”

Figure 5: The details on biological replicates used for microarray dataset related to arbovirus infected mosquitoes are not provided in the referred publication Colpitts et al., 2011. They have only provided the replicate details for the qPCR data where they have used 2 – 3 experimental sets. We even approached the author Dr Tonya Colpitts (Colpitts et al., 2011) for the same, however, we did not receive any response.

Figures 4 and 6 were the qPCR results of the experiments performed in our lab and the details of biological replicates have now been added in figure legends. It should be read as-

“Figure 4. qPCR analysis of randomly selected ABC transporters in different developmental stages of Ae. aegypti mosquito. E= Eggs, L= Larvae (numbers indicate the instar of larvae L1 to L4), P= Pupae, M= Males and F= Females. All samples were collected from three biological replicates at each developmental stage, error bars indicate mean ±SD (n=3). The relative mRNA levels were calculated as mentioned in Materials and Methods. The mRNA levels in eggs were considered as 1.0 in each analysis. Significant differences among relative mRNA levels are represented with different alphabets like “a, b, c”. The ‘a’ alphabet bearing bars have non-significant expression change to each other, while they have significant differences with bars bearing other alphabets (b & c).”

Figure 6. the details of biological replicates were already mentioned in the previous version itself in the figure legend. It should be read as-

“Figure 6. Comparative analysis of the microarray and qPCR expression data of a few representative ABC transporter genes (as mentioned on the x-axis) during DENV2 infection. The fold change in mRNA expression was calculated by ΔΔCt method as described in Materials and Methods, using the actin gene as an internal control. All samples were collected from three biological replicates of each treatment (DENV2 infected blood- v/s uninfected blood-fed) at specified time intervals. Error bars indicate mean ±SD (n = 3). D1, D2 and D7 indicate the days post-viral infection.”

  • Figure 5.  Were all the ABC transporter gene expression data normalized to uninfected (only blood fed) mosquitoes? Please make it clear in the figure legend. It would be better that the viral titer and infection prevalence were included in the figures, and confirmed mosquitoes were infected by YFV, WNV or DENV in different tissues at different days post viral infection.

Answer- In figure 5 we have analyzed the microarray data retrieved from Colpitts et al., 2011 [reference no. 7 in the manuscript]. Therefore, the experimental details were not provided initially. However, following the reviewer’s suggestion, now we have incorporated the same in the legend of figure 5 as mentioned below-

Figure 5. Analysis of Ae. aegypti ABC transporters expression profile in WNV 2741, DENV-2 New Guinea C or YFV Asibi strain infected mosquitoes. As described by Colpitts et al., 2011, virus was used at 6.5 logs per ml, and  0.5 µl  was injected intra-thoracically per mosquito. The viral infection was confirmed by RT-qPCR [7]. (A) The Venn diagram represents the number of exclusively or commonly expressed genes post arboviral infection. The list of these genes is given at the bottom of the diagram. (B) Heatmap of log2 values of differentially regulated transcripts of Ae. aegypti ABC transporters at different days post virus infection [7]. Only the genes exhibiting a significant expression (p<0.05) are represented here. The Color scheme indicates the log2 Fold-change expression of genes against controls at D1, D2, D7 (D=represents the day post-infection).

  • Figure 6. Was any statistical analysis performed on these data? Please indicate.

Answer- This was the comparison of gene expression trends between microarray and qPCR data. Therefore, no statistical analysis was used here.

  • Line 533. Please write details about viral infection. Which methods were used to detect virus titers in mosquitoes? WNV and YFV information was missing here. Please indicate the virus origin and other details.

Answer- This information has already been incorporated in the legend of figure 5 in the revised manuscript as mentioned above.

Reviewer 2 Report

The authors annotate the ABC transporter family in the recently completed Aedes aegypti L5 assembly and characterize expression across several published datasets. This is a useful resource for the vector biology field and can help guide future studies of vector responses to viral infection, providing stable and sensible names for members of the gene family and demonstrating that they are good candidates for further investigation.

For your phylogenetic analysis (L507) — I am not sure that neighbor-joining is sufficient for reconstructing the evolutionary history of a large gene family. At the very least I think maximum likelihood is necessary (e.g. PhyML), or ideally a Bayesian method that also takes into account alignment uncertainty in large and divergent gene families (e.g. BAli-Phy).

In addition, when we were annotating the chemoreceptor gene families for this assembly, we noticed some real genes were not actually in the official gene set or were misassembled (see supplementary materials of the Aedes aegypti L5 genome paper). We resolved this by investigating regions of the genome that showed homology to the gene family but lacked annotated genes, and by manually inspecting regions of TBLASTN homology in and around the gene models themselves. Maybe the same problems don’t exist for the ABC gene family, but it would be good to check and to state in the manuscript that you checked, since this is a known potential issue with annotation.

minor comments:

L102: “~100%” why approximately? maybe just state that they all showed evidence of expression in at least one of the surveyed RNA-Seq datasets?

Figure 1: It would be useful to have the structural topology for the individual transporters indicated in Table 1. Also, Panel 1C is a bit difficult to visually parse — you might consider reducing the size of the chromosomes and increasing the size of the gene labels or something like that?

Figure 3: the clustering analysis looks a bit funny to me in its grouping — did you use Euclidean distance? If so your results may be dominated by absolute level of expression rather than developmental profile per se. You might consider using correlation distance or a similar approach to ensure that clusters are formed of genes with similar developmental expression profiles regardless of absolute expression level.

Figure 4: caption should clarify that this is now qPCR data — clear from the main text but confusing if you go straight to the figure

Figure 5 and section 2.5: “significant expression” is a bit confusing as a term — I believe you mean significantly differentially expressed (p<0.05) relative to controls/baseline — it would be good to spell it out early on and then maybe say “differentially expressed” or DE from there onward?

L274: I think you mean “in boxes”

L364: “a few” -- also, why just a few? would be good to have a supplementary figure or table showing results for all genes (since the goal is to have a comprehensively annotated resource)

L378: something off with grammar, extra “an” perhaps?

L421: should be “performed an in-depth”

Overall, this is a useful and interesting study, and I think it will be a valuable resource for the vector biology community. Thanks to the authors, and don’t hesitate to contact me for any clarification.

Author Response

Comments and Suggestions for Authors

Reviewer 2

The authors annotate the ABC transporter family in the recently completed Aedes aegypti L5 assembly and characterize expression across several published datasets. This is a useful resource for the vector biology field and can help guide future studies of vector responses to viral infection, providing stable and sensible names for members of the gene family and demonstrating that they are good candidates for further investigation.

  • For your phylogenetic analysis (L507) — I am not sure that neighbor-joining is sufficient for reconstructing the evolutionary history of a large gene family. At the very least I think maximum likelihood is necessary (e.g. PhyML), or ideally a Bayesian method that also takes into account alignment uncertainty in large and divergent gene families (e.g. BAli-Phy).

Answer- The topologies of the phylogenetic trees was obtained by both maximum likelihood (ML) and neighbor-joining (NJ) methods and similar results were obtained, therefore, only the NJ based analyses are presented in the manuscript.

  • In addition, when we were annotating the chemoreceptor gene families for this assembly, we noticed some real genes were not actually in the official gene set or were misassembled (see supplementary materials of the Aedes aegypti L5 genome paper). We resolved this by investigating regions of the genome that showed homology to the gene family but lacked annotated genes, and by manually inspecting regions of TBLASTN homology in and around the gene models themselves. Maybe the same problems don’t exist for the ABC gene family, but it would be good to check and to state in the manuscript that you checked, since this is a known potential issue with annotation.

Answer- As suggested by the reviewer, we performed tBLASTn search against Ae. aegypti genome assembly 5 (AaegL5) and obtained similar results as BLASTp analyses. We have updated this information in Materials and Methods of the revised manuscript, and it should read as –

“In addition, tBLASTn search was also performed using the protein sequences of An. gambiae ABC transporters to rule out the possibility of mis/un-annotated or new ABC transporter genes in Ae. aegypti genome assembly 5 (AaegL5).”.

Minor comments: 

  • L102: “~100%” why approximately? maybe just state that they all showed evidence of expression in at least one of the surveyed RNA-Seq datasets?

Answer- As per the reviewer’s suggestion we have reframed the sentence as - “they all showed evidence of expression in at least one of the analyzed transcriptomic datasets”.

  • Figure 1: It would be useful to have the structural topology for the individual transporters indicated in Table 1. Also, Panel 1C is a bit difficult to visually parse — you might consider reducing the size of the chromosomes and increasing the size of the gene labels or something like that?

Answer- The structural topology of all ABC transporters mentioned in Table 1 is already given in figure 1A, Similar structure bearing members of an ABC transporter subfamily was depicted by one architecture and their total numbers are denoted in parenthesis. .

As per reviewers’ suggestions, we have increased the size of gene labels in Figure 1C for better visibility. We are also providing Figure 1A, 1B and 1C as separate figures for better understanding and clarity.

  • Figure 3: the clustering analysis looks a bit funny to me in its grouping — did you use Euclidean distance? If so your results may be dominated by absolute level of expression rather than developmental profile per se. You might consider using correlation distance or a similar approach to ensure that clusters are formed of genes with similar developmental expression profiles regardless of absolute expression level.

Answer- Initially we used hierarchal clustering with the Pearson distance method for the analysis. However, some of the genes had no expression (Zero Value) and since this method cannot calculate the distance between the rows (genes) if all columns (samples) have zero expression (or similar value), therefore, we could not use this method for clustering. We tried performing the same analysis for the genes with non-zero values (51 genes) across the samples. However, the heatmap looked chaotic. Thus, we moved to k-means clustering based on the understanding that “Euclidean distance, Pearson correlation and ‘uncentered’ correlation (angular separation) all seem to work reasonably well as distance measures.” (“D'haeseleer, Patrik. "How does gene expression clustering work?." Nature biotechnology 23.12 (2005): 1499-1501”).

  • Figure 4: caption should clarify that this is now qPCR data — clear from the main text but confusing if you go straight to the figure

Answer- Considering the reviewer's suggestion we have reframed the sentence. Now it should be read as –

Figure 4. qPCR analysis of randomly selected ABC transporters in different developmental stages of Ae. aegypti mosquito”.

  • Figure 5 and section 2.5: “significant expression” is a bit confusing as a term — I believe you mean significantly differentially expressed (p<0.05) relative to controls/baseline — it would be good to spell it out early on and then maybe say “differentially expressed” or DE from there onward?

Answer- As per the reviewer's suggestion, we have reframed the sentences.

  • L274: I think you mean “in boxes”

Answer- As per the reviewer's suggestion, it is corrected and should read as ‘in boxes’.

  • L364: “a few” -- also, why just a few? would be good to have a supplementary figure or table showing results for all genes (since the goal is to have a comprehensively annotated resource)

Answer- Our major focus for this section was on the immune regulation of mosquito ABC transporters during arbovirus infection. Therefore, we have included representative virus-induced transporter genes and identified the major immunity-related TFs in their 5’ UTR region (Figure 7A). We have also provided a table in the Supplementary section (Supplementary Table 2) detailing the function of these TFs. The total number of putative TFs was very high for each ABC transporter, therefore to concise the data, we have included the representatives only.

  • L378: something off with grammar, extra “an” perhaps?

 Answer- We thank the reviewer and the sentence has been reframed.

  • L421: should be “performed an in-depth”

Answer- As per the reviewer's suggestion, the sentence has been corrected.

Overall, this is a useful and interesting study, and I think it will be a valuable resource for the vector biology community. Thanks to the authors, and don’t hesitate to contact me for any clarification.

This manuscript is a resubmission of an earlier submission. The following is a list of the peer review reports and author responses from that submission.

Round 1

Reviewer 1 Report

General comments:

- Figure legends are still incomplete and do not explain acronyms.

- Virus acronyms are not explained in abstract and explained many times during text.

- Normalization does not seem to have been corrected in the comparison between microarray and RNA-seq analysis.

- Obscure changes in results (ex. figure 6 and figure 8) were not explained.

Reviewer 2 Report

In this work, Kumar and colleagues performed an interesting study comprehensively identifying ABC transporter genes in Ae. aegypti genome and assessing its transcriptional profile along with different arbovirus-infection conditions and developmental stages.

Although I recognize the hard work the authors performed with biological validations and complex bioinformatics analyses, there are a few concerns that authors should work out before the manuscript is suitable for publication.

Minor concerns

Line 27-28. In organisms from the different domains of life

Line 55-57. What are the References?

Line 62-64. RNAi can also cleave the RNA genome.

Line 71. ABCC ?

Figure 1B is too small. The authors have space to enlarge the figure and figure labels in order to make it clearer.

What about ABCE1 and ABCC17 grouping in figure 1?

The authors could highlight the genes analyzed by qPCR in the heatmap in Figure 3. It helps to identify those elements further analyzed.

Major concerns

What about the similarity among Ae. Aegypti ABC transporters ? It could lead to false positives in expression profiling analysis since identical fragments identified in different proteins associated with multiple mapping could mislead the results. In addition, the parameters and software used for the remapping were not provided.

What a,b and c mean in Figure 4. There are no details on the figure legend which impairs the correct interpretation of the figure.

Authors should enlight the comparisons performed for differential expression analysis showed in figure 5... All of them were performed infected VS control?

Where is the full list of TFs predicted for the genes? It should be supplied as Sup. Table.

Figure 8 does not make sense without correlation with the expression profiles observed in control VS arboviruses infected individuals since most to the TF are related to immunity.